# Child–Pugh Class and Not Thrombocytopenia Impacts the Risk of Complications of Endoscopic Band Ligation in Patients with Cirrhosis and High Risk Varices

**DOI:** 10.3390/jpm13050764

**Published:** 2023-04-28

**Authors:** Vincenzo Di Martino, Fabio Simone, Maria Grasso, Yasmin Abdel-Hadi, Marco Peralta, Marzia Veneziano, Antonino Lombardo, Sergio Peralta, Vincenza Calvaruso

**Affiliations:** Section of Gastroenterology and Hepatology, Department of Health Promotion, Mother and Child Care, Internal Medicine and Medical Specialties (PROMISE), University of Palermo, Piazza delle Cliniche n.2, 90127 Palermo, Italy

**Keywords:** variceal bleeding (ABV), esophageal band ligation (EBL), cirrhosis, platelets, thrombocytopenia, Child–Pugh score, MELD score, clinically significative portal hypertension (CSPH)

## Abstract

**Background and Aims:** Endoscopic band legation (EBL) is an effective method for the prophylaxis of acute variceal bleeding (AVB). This procedure may be associated with several complications, particularly bleeding. Our analysis aimed to evaluate the risk of complications due to EBL in a cohort of patients who underwent EBL for the prophylaxis of variceal bleeding and the eventual presence of risk predictors. **Patients and Methods:** We retrospectively analysed data from consecutive patients who underwent EBL in a primary prophylaxis regimen. For all patients, simultaneously with EBL, we recorded the Child–Pugh and MELD score, platelet count and US features of portal hypertension. **Results:** We collected data from 431 patients who performed a total of 1028 EBLs. We recorded 86 events (8.4% of all procedures). Bleeding after EBL occurred 64 times (6.2% of all procedures), with the following distribution: intraprocedural bleeding in 4%; hematocystis formation in 17 cases (1.7%); 6 events (0.6%) of AVB due to post-EBL ulcers. None of these events presented a correlation with platelet count (84,235 ± 54,175 × 10^3^/mL vs. 77,804 ± 75,949 × 10^3^/mL; *p* = 0.70) or with the condition of severe thrombocitopenia established at PLT < 50,000/mmc (22.7% with PLT ≤ 50,000/mmc vs. 15.9% with PLT ≥ 50,000/mmc; *p* = 0.39). Our results showed a relationship between cumulative complications of EBL and Child–Pugh score (6.9 ± 1.6 vs. 6.5 ± 1.3; *p* = 0.043). **Conclusions:** EBL in cirrhotic patients is a safe procedure. The risk of adverse events depends on the severity of liver disease, without a relationship with platelet count.

## 1. Introduction

Acute variceal bleeding (AVB) is one of the most common complications in patients with liver cirrhosis, with a mortality of around 20% and an increasing risk of new events of bleeding in the absence of prevention measures [1,2]. Thus, patients with medium–large varices start a primary prophylaxis from bleeding episodes with endoscopic band ligation (EBL) or the administration of nonselective beta-blockers (propranolol, carvedilol, nadolol). EBL is an option for patients who do not tolerate beta-blockers or have contraindications to those drugs [3]. In patients that show an inadequate reduction in portal pressure at HVPG, EBL is performed together with the administration of nonselective beta-blockers. The most common and feared complication of EBL is bleeding. This may occur during the procedure or after 7–21 days after the EBL, due to post-EBL ulcers, with an incidence of 1.5–10% [4,5]. Portal hypertension and advanced chronic liver disease with high Child–Pugh and elevated model for end-stage liver disease (MELD) score are identified as risk factors of post-EBL bleeding [4]. Conventional tests that evaluate coagulation, such as prothrombin time (PT), international normalized ratio for prothrombin time (INR) and activated partial thromboplastin time (aPTT) fail to predict bleeding risk in patients who undergo EBL. Limited data support the evidence of a higher risk of post-EBL bleeding in the presence of extremely low platelet count (≤50,000/mmc) [6]. Furthermore, platelet count increases for a limited period after platelet apheresis, while bleeding often occurs 1–2 weeks after EBL. Thus, the administration of platelet apheresis in patients with an extremely low platelet count before EBL is not standardized. The administration of platelets may by complicated by infections and anaphylactic reactions and also increase the risk of bleeding due to portal pressure elevation from excessive volume expansion [6,7]. Data on the effectiveness and safety of new TPO-receptors agonists are still limited.

This monocentric retrospective observational study evaluates the global incidence of adverse events that may occur after EBL and, in particular, the incidence of bleeding. It also aimed to investigate the possible risk factors.

## 2. Patients and Methods

We retrospectively reviewed data of consecutive inpatients and outpatients who performed EBL for primary prophylaxis at Gastroenterology Unit of the AOUP Paolo Giaccone of Palermo between January 2002 and December 2021. We excluded patients treated for secondary prophylaxis and those who presented with active variceal bleeding at admission. All patient data on age, sex, liver disease aetiology (HBV, HCV, alcohol, NASH, autoimmune, other) were collected, as well as haemoglobin, platelet count, creatinine, bilirubin and albumin. Child–Pugh score and model for end-stage liver disease (MELD) score were also evaluated. We performed abdominal US to evaluate portal hypertension signs, such as portal vein diameter, the presence or absence of portal vein thrombosis, spleen longitudinal diameter and the presence of ascites or pleural effusion. Additionally, the presence of HCC at abdominal US was detected. Finally, we recorded data from antiplatelet therapy and anticoagulant therapy with heparin, warfarin or direct-acting oral anti-coagulant (DOAC). Varices were classified according to AASLD guidelines. Esophageal varices (EV) were classified for dimension in relation to luminal maximal occupation: F1 < 33%; F2 between 33% and 66%; F3 > 66%. Endoscopists described the presence of high-risk bleeding signs, such as red spots. Gastric varices were classified as follows: GOV1 for esophagogastric varices that extends along the gastric lesser curvature; GOV2 for esophagogastric varices that extends toward the fundus; IGV1 as isolated varices in gastric fundus; and IGV2 as varices allocated elsewhere in the stomach. EBL was performed by expert endoscopists after a thorough diagnostic EGD according to ESGE guidelines and with the application of a maximum of six bands from the same device (Cook Medical). All patients received intravenous sedation with Midazolam. According to the guidelines, patients discontinued antiplatelet therapy 5 days before EBL, heparine 24 h before EBL and warfarin or direct-acting oral anti-coagulant (DOAC) 48 h before EBL. Oral food intake was forbidden for a period of 12 h after the procedure and all patients received a paper with diet advice to apply for a month after EBL. For all patients, a new EGD with eventual EBL was programmed after 1 month from the previous. Administration of proton pump inhibitors (PPIs) and analgesics occurred at physician’s discretion. We consider eradication to be achieved when the varices become too small for further banding ligation. Bleeding was defined as direct observation of blood dripping during EBL, hematocystic spots’ formation or any episode of hematemesis and/or melena after 1–2 weeks by EBL. Any post-EBL bleeding was managed according to guidelines. Other adverse events were represented by substance-loss ulcers, strictures, sepsis, epigastralgia and glottis oedema.

## 3. Statistical Analysis

Data for continuous variables were expressed as mean and standard deviation (SD) or median and interquartile ranges (IQR), and data for categorical variables were expressed as frequency and percentage. Differences between continuous data were assessed by Student *t* test or by Mann–Whitney U test. Differences between categorical variables were assessed by χ^2^ test. A *p* value < 0.05 represented statistical significance.

Analysed outcomes were the intraprocedural bleeding, hematocystic spots’ formation or any episode of AVB due to post-EBL ulcer. Secondary outcome was the occurrence of other complications related to EBL procedure. The analysis was conducted with SPSS system.

## 4. Results

We reviewed data from 431 patients that underwent 1028 procedures between January 2002 and December 2021. A total of 287 patients (66.6%) obtained variceal eradication with a per-patient procedure mean of 2.23 (±1.17), within a mean time of 8.93 months. The baseline characteristics are shown in Table 1, while Table 2 shows the varices’ baseline dimensions and characteristics.

Eighty-six events (8.4% of all procedures) were recorded. Details of these events and their percentage are shown in Table 3. Bleeding occurred 64 times (6.2% on all procedures) with the following distribution: intraprocedural bleeding in 4%, 6 of all (14.6%), with further need of vasoactive drugs’ administration; 17 episodes (1.7%) of hematocystic spots formation; 6 events (0.6%) of AVB due to post-EBL ulcers.

Table 4 describes the cases of intraprocedural bleeding. These occurred in 16 patient with F2 varices (39%) and in 25 patients (61%) with F3 varices. No significant differences were found in relation to varices’ size (*p* = 0.5). Mean platelet count in patients with intraprocedural bleeding was 80,948 ± 37,658 × 10^3^/mL, compared to 87,837 ± 75,729 × 10^3^/mL in patients who did not develop this event (*p* = 0.5). A total of 7 patients (17.0%) with an intraprocedural bleeding had a platelet count ≤50,000/mmc, but 162 patients with severe thrombocytopenia (16.4%) did not bleed during the procedure (*p* = 0.6). Considering the total number of procedures, 169 EBLs (16.4%) were performed in patients with a platelet count ≤50,000/mmc and intraprocedural bleeding occurred in 7 procedures (4%); 859 procedures (83.6%) were performed in patients with a platelet count >50,000/mmc, with an incidence of intraprocedural bleeding of 3.7%. In patients that bled during EBL, the mean number of bands was similar to the mean bands in patients who did not bleed during the procedure (5.1 ± 1.3 vs. 5.2 ± 1.5; *p* = 0.7). There was no significant association between intraprocedural bleeding and the use of antithrombotic drugs, *p* = 0.7. Patients that presented with bleeding during EBL had the same Child–Pugh score (6.8 ± 1.8 vs. 6.5 ± 1.5; *p* = 0.3) and MELD score (8.3 ± 4.7 vs. 8.4 ± 3.4; *p* = 0.8) as patients without intraprocedural bleeding.

In Table 5, we summarize data on hematocystic spots’ formation. The mean platelet count was not different in patients with hematocystic spots’ formation and patients that did not develop this event (79,0250 ± 44,713 × 10^3^/mL vs. 87,170 ± 74,825 × 10^3^/mL; *p* = 0.65). Among patients with a platelet count <50,000/mmc, 5 (29.4%) developed hematocystic spots, compared to 164 (16.2%) without hematocystic spots’ formation (*p* = 0.3). The mean number of bands was the same in patients with hematocystic spots’ formation and in patients without this complication (5.3 ± 1.1 vs. 5.2 ± 1.5; *p* = 0.74). There was no significant association between hematocystic spots’ formation and the use of antithrombotic drugs, *p* = 0.5. We also did not find differences in Child–Pugh score (6.8 ± 1.1 vs. 6.5 ± 1.3; *p* = 0.46) and MELD score (8.4 ± 2.8 vs. 8.4 ± 3.5; *p* = 0.98).

Table 6 shows the results regarding AVB due to post-EBL ulcers. No differences were found in the occurrence of this event due to varices’ size (*p* = 0.2). The mean platelet count was not different in patients with bleeding due to post-EBL ulcers and patients that showed this event (125,666 ± 112,348 × 10^3^/mL vs. 87,217 ± 74,011 × 10^3^/mL; *p* = 0.44). One patient (16.7%) who developed AVB due to post-EBL falling ulcer had an extremely low platelet count, while the number of patients with a platelet count <50,000/mmc and no AVB was 168 (*p* = 0.08). the mean number of bands was similar both in patients with AVB after 1–2 weeks and in patients in whom this event did not occur (5.8 ± 1.6 vs. 5.2 ± 1.5; *p* = 0.29). No cases of AVB due to post-EBL ulcers occurred in patients who were treated with antithrombotic drugs. Child–Pugh score (6.8 ± 1.6 vs. 6.5 ± 1.4; *p* = 0.74) and MELD score (8.5 ± 3 vs. 8.4 ± 3.5; *p* = 0.95) were not dissimilar in the two groups.

Table 7 shows a cumulative analysis of all the recorded events. Child–Pugh score was the only feature with a marginal relationship with the risk of events after EBL (6.9 ± 1.6 in patients who experienced an event vs. 6.5 ± 1.3 in patients who did not; *p* = 0.043). Instead, mean platelet count (84,235 ± 54,175 × 10^3^/mL vs. 77,804 ± 75,949 × 10^3^/mL; *p* = 0.70) and severe thrombocitopenia (22.7% with plt < 50,000/mmc vs. 15.9% with plt ≥ 50,000/mmc; *p* = 0.39) did not appear significantly different in patients who developed any event and those who did not.

Analysing the cumulative events, we did not find a significant association between EBL complications and the use of antithrombotic drugs, *p* = 0.7.

## 5. Discussion

Our study evaluates the global incidence of adverse events in cirrhotic patients who performed EBL and investigates their probable predictors. The cumulative incidence of adverse events in our population is similar to the percentage that was described in previous studies [8,9,10]. Our results confirm intraprocedural bleeding as the most common adverse event in patients who perform EBL. Regarding bleeding, described as intraprocedural bleeding, hematocystis formation or bleeding due to post-EBL ulcers, we evaluated if its occurrence could be influenced by platelet count during EBL. In fact, patients with liver cirrhosis often present with a low platelet count that reflects the splenomegaly due to portal hypertension and the altered hepatic synthesis of thrombopoietin. Our analysis concludes that there is not a correlation between any form of bleeding and both mean platelet count and the presence of a severe thrombocitopenia (plt < 50,000/mmc). These results are in line with the latest guidelines about procedural bleeding, which do not identify a platelet, INR and fibrinogen cut-off as useful to predict bleeding risk during several procedures [6,11]. Thus, our data support the hypothesis that cirrhosis is not only associated with the risk of bleeding, but determines a complex modification of the global haemostasis. This phenomenon results in an unpredictable risk both of bleeding and clotting. The new blood viscoelastic tests, such as thromboelastography (TEG) and rotational thromboelastometry (ROTEM), are promising in this setting, but do not the ability to predict bleeding at present [12,13]. Cumulative events’ analysis shows that Child–Pugh score measured before EBL is the only variable that reaches statistical significance. This result suggests that liver function and portal hypertension are the main drivers of adverse events, particularly bleeding. Our results confirm the conclusions of the study recently performed by Blasi A et al. [14], who demonstrated that the incidence of post-EBL bleeding is associated with advanced liver disease and there is no association between INR or platelet count and bleeding events.

The current study presents some limitations: first of all, it is a retrospective analysis, and thus a prospective and longer assessment of the adverse events after EBL was not performed; second, we did not include patients undergoing secondary prophylaxis and with active bleeding at admission due to the heterogeneity of the factors involved in these settings (i.e., vasoactive therapy, patients’ conditions); finally, although there was a wide sample size, the study was limited to a single centre.

In conclusion, our study proves that EBL in cirrhotic patients is associated with a very low incidence of adverse events, confirming the good safety of the procedure. The risk of adverse events depends on portal hypertension and the stage of liver disease. There is no any association between procedural bleeding and platelet count.

## Figures and Tables

**Table 1 jpm-13-00764-t001:** Demographic and clinical characteristics of 431 patients with cirrhosis and HRV who underwent at least one EBL procedure.

	Patients (Tot: 431)
**Age, y ± SD**	62.7 ± 12.7
**Sex M, *n* (%)**	281 (65.2%)
**INR ± SD**	1.2 ± 0.3
**Bilirubin mg/dL ± SD**	1.5 ± 1.5
**Albumin g/dL ± SD**	3.3 ± 0.6
**Creatinie mg/dL**	0.7 ± 0.6
**Platelets/mm^3^**	92,272 ± 83,320
**Child–Pugh** (median)	6
A	281 (65.2%)
B	134 (31.1%)
C	16 (3.7%)
**Etiology *n* (%)**	
Viral	238 (55.2)
- ASH	40 (9.3)
- NASH	61 (14.2)
- Autoimmune hepatitis	15 (3.5)
- ASH + viral	7 (1.6)
Other	70 (16.2)
**Portal vein diameter, mm ± SD**	13.0 ± 0.85
**Spleen LD, cm ± SD**	16.1 ± 0.7
**- Liver events, *n* (%)**	
- Ascites *n* (%)	183 (42.5)
- Mild	62 (14.4)
- Moderate	82 (19)
- Severe	39 (9)
Pleural effusion *n* (%)	22 (5)
- HE *n* (%)	27 (6.3)
- Portal vein thrombosis *n* (%)	72 (16.7)
**HCC *n* (%)**	73 (17)
**Antithrombotic therapy**	38 (8.8)
Antiplatelets (ASA or cropidogrel)	13 (3.0)
LMWH	19 (4.4)
Warfarin	6 (1.4)

ASH: alcoholic steatohepatitis; NASH: non-alcoholic steatohepatitis; HCC: hepatocellular carcinoma; LMWH: low-molecular-weight heparin; HE: hepatic encephalopathy.

**Table 2 jpm-13-00764-t002:** Varices dimensions and characteristics of 431 patients with cirrhosis and HRV who underwent at least one EBL procedure.

Esophageal Varices *n* (%)	
- F2	130 (30)
- *(F2 occ—dim*, *median)*	*40–40*
- F2 red marks	48 (11)
- *(F2 red marks occ—dim*, *median)*	*40–50*
- F3	81 (18.8)
- *(F3 occ—dim*, *median)*	*60–70*
- F3 red marks	172 (39.9)
- *(F3 red marks occ—dim*, *median)*	*70–80*
**Gastric varices *n* (%)**	76 (17.6)
Gov1	47 (10.9)
Gov2	15 (3.5)
Igv1	9 (2)
Igv2	3 (0.7)
Other GV	5 (1.2)

**Table 3 jpm-13-00764-t003:** Features of complications of EBL that occurred in 431 patients with cirrhosis and HRV.

Complications	EBL Number (%)
**Total**	**86 (8.4)**
**Intraprocedural bleeding**	**41(4)**
**Need for intervention**	**6 (14.6)**
**Hematocystic spots’ formation**	**17 (1.7)**
**Bleeding due to post-EBL ulcers**	**6 (0.6)**
**Esophageal ulcers**	**11 (1.1)**
**Strictures**	**1 (0.1)**
**Sepsis**	**2 (0.2)**
**Epigastralgia**	**7 (0.7)**
**Glottal edema**	**1 (0.1)**

EBL: endoscopic band ligation.

**Table 4 jpm-13-00764-t004:** Risk factors for intraprocedural bleeding during prophylactic endoscopic variceal band ligation.

	Intraprocedural Bleeding (41)	No Bleeding (987)	*p* Value
EV *n*(%)			0.5
F2	16 (39)	389 (39.4)	
F3	25 (61)	598 (60.6)	
PLT mean	80,948.00 ± 37.658	87,837.00 ± 75.729	0.5
PLT ≥ 50,000	34 (83)	825 (83.6)	
PLT < 50,000	7 (17.0)	162 (16.4)	0.6
Bands N	5.1 ± 1.3	5.2 ± 1.5	0.7
Antithrombotic drugs	2 (4.9)	36 (3.6)	0.7
MELD	8.3 ± 4.7	8.4 ± 3.4	0.8
CP	6.8 ± 1.8	6.5 ± 1.5	0.3

**Table 5 jpm-13-00764-t005:** Risk factors for hematocystic spots’ formation during prophylactic endoscopic variceal band ligation.

	Hematocystic Spots (17)	No Hematocystic Spots (1011)	*p* Value
EV *n* (%)			0.019
F2	12 (70)	394 (39)	
F3	5 (30)	617 (61)	
Mean PLT ± SD	79,250.00 ± 44,713	87,170 ± 74,825	0.7
PLT < 50,000	5 (29.4%)	164 (16.2%)	0.3
N bands ± SD	5.3 ± 1.1	5.2 ± 1.5	0.7
Antithrombotic drugs	1 (5.9)	37 (3.7)	0.5
MELD ± SD	8.4 ± 2.8	8.4 ± 3.5	0.9
CP ± SD	6.8 ± 1.1	6.5 ± 1.3	0.5

**Table 6 jpm-13-00764-t006:** Risk factors for band-induced ulcer bleeding after prophylactic endoscopic variceal band ligation.

	Bleeding Due to Post-EBL Ulcers (6)	No Bleeding Due to Post-EBL Ulcers (1022)	*p* Value
EV *n* (%)			0.2
F2	4 (66.6)	402 (39.3)	
F3	2 (33.4)	620 (60.7)	
PLT mean ± SD	125,666.00 ew ± 112,384	87,217 ± 74,011	0.4
PLT < 50.000	1 (16.7)	168 (16.4)	0.08
Bands N ± SD	5.8 ± 1.6	5.2 ± 1.5	0.3
Antithrombotic drugs	0	38 (3.7)	0.9
MELD ± SD	8.5 ± 3	8.4 ± 3.5	0.9
CP ± SD	6.8 ± 1.6	6.5 ± 1.4	0.73

**Table 7 jpm-13-00764-t007:** Risk factors and cumulative complications after prophylactic endoscopic variceal band ligation.

	Cumulative Events (75)	No Events (953)	*p* Value
EV *n* (%)			0.2
F2	35 (46.7)	372 (39)	
F3	40 (53.3)	581 (61)	
PLT mean ± DS	84,235 ± 54,175	77,804 ± 75,949	0.7
PLT < 50,000 pl	17 (22.7%)	152 (15.9%)	0.4
Bands N. ± SD	5.1 ± 1.6	5.2 ± 1.5	0.8
Antithrombotic drugs	3 (4)	35 (3.7)	0.7
MELD ± SD	8.5 ± 4.2	8.4 ± 3.4	0.9
CP ± SD	6.9 ± 1.6	6.5 ± 1.3	**0.043**

## Data Availability

Dataset with data supporting reported results are available.

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
