# Peer review of "Child–Pugh Class and Not Thrombocytopenia Impacts the Risk of Complications of Endoscopic Band Ligation in Patients with Cirrhosis and High Risk Varices"

_jpm, 2023, doi:10.3390/jpm13050764_

Round 1

Reviewer 1 Report

This is an interesting topic and in the manuscript results of profylactic esophageal band ligation (EBL) in patient with liver cirrhosis are described. Differences between patients with intraprocedural bleeding and no bleeding are analysed. 

The cohort is fairly large, the topic is of interest as well as results.

Although the material is interesting, it is not presented in an adequate way. The authors should follow the STROBE guidelines.

In methods all parameters should be described or refered to, including definitions of varices (Gov 1 , F3 etc)

Results should be described in a better way. Results are not in alignment with aims. The manuscript contains several results where the hypothesis or research question is unclear.

Statistics are very basic and no adjustment for confounders are made. Risk factors for bleeding are not analysed with appropriate statistic methods. Subgroup analysis should not be done since number of patients are too small in some groups (ex F2 in table 4 and 5 is n=1)- instead larger groups should be compared. There might be type 2 errors with the current design.

Known or possible confounders as acute and chronic kidney disease, venous thromboembolism (VTE) prophylaxis, DOACs and others should be described and be adjusted for. 

Lots of abbreviations (HSV, APS, EPV, HVPG and many more) are unexplained, both in tables and in text. They should be written out first time used and in table text. All abbreviations are not mentioned in List of abbreviations. All tables should be possible to read without reading the text. 

Potential biases discussed should be discussed in discussion.  

Minor things: is Hematocystis-the correct term? Should it be hematocystic spots? There is no rationale explaining why this is analysed.

References (ex ref 11) are not correct, and several references are not formatted correct.

Author Response

This is an interesting topic and in the manuscript results of prophylactic esophageal band ligation (EBL) in patient with liver cirrhosis are described. Differences between patients with intraprocedural bleeding and no bleeding are analysed. 

The cohort is fairly large, the topic is of interest as well as results.

Although the material is interesting, it is not presented in an adequate way. The authors should follow the STROBE guidelines.  In methods all parameters should be described or refered to, including definitions of varices (Gov 1, F3 etc).

Results should be described in a better way. Results are not in alignment with aims. The manuscript contains several results where the hypothesis or research question is unclear.

R: We thank the reviewer for the suggestion, we have now better described the variables and inserted the complete definition of varices as suggested. We have also better clarified the outcomes analysed and the aim of the study.

Statistics are very basic and no adjustment for confounders are made. Risk factors for bleeding are not analysed with appropriate statistic methods. Subgroup analysis should not be done since number of patients are too small in some groups (ex F2 in table 4 and 5 is n=1)- instead larger groups should be compared. There might be type 2 errors with the current design.

R: We have now performed the analysis cumulating the group of patients according to varices size in order to compare larger groups.

Known or possible confounders as acute and chronic kidney disease, venous thromboembolism (VTE) prophylaxis, DOACs and others should be described and be adjusted for. 

R: We have evaluated the comorbidity of the patients analysed. Eleven patients had a diagnosis of chronic kidney disease. Thirty-eight patients were treated with antithrombotic drugs, 19 with Low molecular weight heparin, 6 with warfarin and 13 with antiplatelets drugs. No patients were treated with DOACs.

We have inserted in the analysis the use of antithrombotic drugs. We have also inserted in the methods the discontinuation rule used for these drugs before EBL. We have not found a significant association between the use of antithrombotic drugs and complications of EBL.

Lots of abbreviations (HSV, APS, EPV, HVPG and many more) are unexplained, both in tables and in text. They should be written out first time used and in table text. All abbreviations are not mentioned in List of abbreviations. All tables should be possible to read without reading the text. 

R: We have now explained all the abbreviation in the tables and in text and we mentioned all abbreviations in the list of abbreviations.

Potential biases discussed should be discussed in discussion.  

Minor things: is Hematocystis-the correct term? Should it be hematocystic spots? There is no rationale explaining why this is analysed.

R: we have changed the term in hematocystic spots. We analysed also this outcome since we considered the hematocystic spots as a condition with an increased risk of variceal hemorrhage

References (ex ref 11) are not correct, and several references are not formatted correct.

R: We have now correctly formatted the references

Reviewer 2 Report

The authors report a large number of cases evaluating the complication risk of elective EBL for esophageal varices.

1) Could a statistically significant difference in Child-Pugh score be a clinically significant difference? Since there is no difference in the MELD score, I think that it is necessary to reconsider whether the difference can be clinically significant, including the conclusion.

2) I think it would be better to present data on the use of antithrombotic drugs as well.

Author Response

The authors report a large number of cases evaluating the complication risk of elective EBL for esophageal varices.

Could a statistically significant difference in Child-Pugh score be a clinically significant difference? Since there is no difference in the MELD score, I think that it is necessary to reconsider whether the difference can be clinically significant, including the conclusion.

R: Child-Pugh score better reflect the liver function class of the whole cohort of patients analysed (mostly in class A and therefore with compensated liver disease).

I think it would be better to present data on the use of antithrombotic drugs as well.

R: We thanks the reviewer for the suggestion. Thirty-eight patients were treated with antithrombotic drugs, 19 with Low molecular weight heparin, 6 with warfarin and 13 with antiplatelets drugs. No patients were treated with DOACs.

We have inserted in the analysis the use of antithrombotic drugs. We have also inserted in the methods the discontinuation rule used for these drugs before EBL. We have not found a significant association between the use of antithrombotic drugs and complications of EBL.

Reviewer 3 Report

This manuscript evaluates the complications of endoscopic bald ligation for esophageal varices in liver function and thrombocytopenia.However, the usefulness and safety of endoscopic bald ligation for esophageal varices have already been reported in several reports, and there is no new evidence in this manuscript.

Author Response

This manuscript evaluates the complications of endoscopic bald ligation for esophageal varices in liver function and thrombocytopenia. However, the usefulness and safety of endoscopic bald ligation for esophageal varices have already been reported in several reports, and there is no new evidence in this manuscript.

R: We agree with reviewer that the safety of endoscopic band ligation has been already established but the main aim of our study is to demonstrate that the low rate of complications are not related to thrombocytopenia and therefore that endoscopic band ligation may be safely performed in patients with platelets lower than 50,000 mmc without platelet transfusion or thrombopoietin receptor agonists.

Round 2

Reviewer 1 Report

Although this is an interesting topic and the cohort is fairly large, the manuscript is not written or prepared in a scientific way. I cannot find that the comments from last are fulfilled on all points.

Methods section has improved but is still not structured and it is not possible to find definitions or references for all datapoints needed (ex definition of ascites in mild, moderate, severe groups). The manuscript contains several results where the hypothesis or research question is unclear.

Statistics are very basic and no adjustment for confounders are made. Risk factors for bleeding are not analysed with appropriate statistic methods. Considering the number of patients with bleeding at least the most important counfounders (such as thrombocytompenia) should be adjusted for in a relevant multivariable statistical analysis.

Some abbreviations (in abstract US and MELD in introduction ex HVPG, HCC etc) are still unexplained, however MELD is described twice in introduction.

Additionally results are hard to evaluate for example:

Table 1 several errors – why median for Child-Pugh?

Table 2 are F2 with red marks included in F2 or does F2 mean F2 without red marks? What does occ- dim, median mean?

Table 3- not possible to read in downloaded manuscript

Reviewer 2 Report

I have no additional comments.

Reviewer 3 Report

This manuscript has been improved and there are no additional comments.